# Controlling $^{229}$Th isomeric state population in a VUV transparent crystal

Takahiro Hiraki [1,9], Koichi Okai[1,9], Michael Bartokos [2], Kjeld Beeks [2], Hiroyuki Fujimoto[3], Yuta Fukunaga[1], Hiromitsu Haba[4], Yoshitaka Kasamatsu [5], Shinji Kitao[6], Adrian Leitner [2], Takahiko Masuda [1], Ming Guan [1], Nobumoto Nagasawa [7], Ryoichiro Ogake[1], Martin Pimon [2], Martin Pressler [2], Noboru Sasao[1], Fabian Schaden [2], Thorsten Schumm [2], Makoto Seto[6], Yudai Shigekawa [4], Kotaro Shimizu[1], Tomas Sikorsky [2], Kenji Tamasaku [8], Sayuri Takatori [1], Tsukasa Watanabe[3], Atsushi Yamaguchi [4], Yoshitaka Yoda[7], Akihiro Yoshimi [1] ✉ & Koji Yoshimura [1]

The radioisotope thorium-229 ($^{229}$Th) is renowned for its extraordinarily low-energy, long-lived nuclear first-excited state. This isomeric state can be excited by vacuum ultraviolet (VUV) lasers and $^{229}$Th has been proposed as a reference transition for ultra-precise nuclear clocks. To assess the feasibility and performance of the nuclear clock concept, time-controlled excitation and depopulation of the $^{229}$Th isomer are imperative. Here we report the population of the $^{229}$Th isomeric state through resonant X-ray pumping and detection of the radiative decay in a VUV transparent $^{229}$Th-doped CaF$_2$ crystal. The decay half-life is measured to 447(25) s, with a transition wavelength of 148.18(42) nm and a radiative decay fraction consistent with unity. Furthermore, we report a new "X-ray quenching" effect which allows to de-populate the isomer on demand and effectively reduce the half-life. Such controlled quenching can be used to significantly speed up the interrogation cycle in future nuclear clock schemes.

The exceptionally low-energy nuclear-excited state of the isotope $^{229}$Th has attracted considerable attention because of its potential use in ultra-precise optical clocks[1–4]. Such nuclear clocks will find multiple applications, ranging from fundamental physics studies[5–7] to practical implementations as compact solid-state metrology devices[8,9]. With only a few electron volts of excitation energy, this laser-accessible first excited state is predicted to have a long half-life exceeding $10^3$ s (called an "isomer"), making it a candidate for a clock state.

Given that direct excitation experiments from the ground to the isomeric state $^{229m}$Th are ongoing worldwide[10,11], a more precise determination of the isomer half-life, and a measure of radiative

decay fraction in material environments is highly desirable. The spectroscopic studies on the isomer transition and the developments of the nuclear clock are currently focusing on two different physical implementations: Th$^{3+}$ in an ion trap[12] and thorium-doped ionic single crystals with a large band gap[8,9]. A valuable feature of the latter "solid-state nuclear clock" using thorium-doped crystals is that a much larger number of thorium atoms ($\approx 10^{18}$ cm$^{-3}$) can be used than in an ion-trap-based clock. This allows the solid-state nuclear clock to be used as the fast-averaging reference clock, wherein the temporal stability converges with a short averaging time.

[1]Research Institute for Interdisciplinary Science, Okayama University, Okayama 700-8530, Japan. [2]Institute for Atomic and Subatomic Physics, TU Wien, Vienna 1020, Austria. [3]National Institute of Advanced Industrial Science and Technology (AIST), 1-1-1 Umezono, Tsukuba, Ibaraki 305-8563, Japan. [4]RIKEN, 2-1 Hirosawa, Wako, Saitama 351-0198, Japan. [5]Graduate School of Science, Osaka University, Toyonaka, Osaka 560-0043, Japan. [6]Institute for Integrated Radiation and Nuclear Science, Kyoto University, Kumatori-cho, Sennan-gun, Osaka 590-0494, Japan. [7]Japan Synchrotron Radiation Research Institute, 1-1-1 Kouto, Sayo-cho, Sayo-gun, Hyogo 679-5198, Japan. [8]RIKEN SPring-8 Center, 1-1-1 Kouto, Sayo-cho, Sayo-gun, Hyogo 679-5148, Japan. [9]These authors contributed equally: Takahiro Hiraki, Koichi Okai. ✉e-mail: yoshimi@okayama-u.ac.jp

The development of a solid-state nuclear clock requires a superior-grade [229]Th-doped crystal, characterized by an adequate concentration, facilitating isomer excitation and detection of the isomeric decay while suppressing non-radiative decay processes in the crystal. Theoretical and experimental studies on thorium-doped crystals such as Th:CaF$_2$ suggest that the thorium substitutes a Ca$^{2+}$ ion in charge state Th$^{4+}$ in the crystal[13,14], realizing [Rn] electronic configuration. The absence of energetically near-by electronic states should suppress the internal conversion of [229m]Th and make direct radiative vacuum ultraviolet (VUV) photon emission the dominating decay process.

One of the crucial matters to verify is whether the excited isomeric state indeed shows the predicted long radiative lifetimes also in the crystal environment. In 2023, a successful optical measurement of the radiative decay of [229m]Th was reported for the first time[15]. The authors implanted the radioisotopes [229]Fr and [229]Ra into MgF$_2$ and CaF$_2$ crystals at the CERN-ISOLDE facility, where the isotopes undergo successive beta-decays to [229]Th with an uncertain branching into [229m]Th. The wavelength of the detected photon from the isomer state was determined to be 148.71(41) nm by a VUV-spectrometer corresponding to an isomer energy of 8.338(24) eV. The half-life of the isomer in the MgF$_2$ crystal was determined to be 670(102) s through the analysis of a time spectrum encompassing multiple beta decays and isomeric decay. If we restrict our discussion to cases where the charge state of thorium is such that it is 4+, i.e., where internal conversion decay of [229m]Th is forbidden, the half-life of [229m]Th in a crystal is expected to be reduced by a factor $n^3$ ($n$ being the refractive index of the crystal at the relevant wavelength) as compared to the vacuum half-life[16,17]. Influences of the shallow implantation depth (17 nm) compared with the wavelength of the VUV photon from [229m]Th, realized in ref. 15, require further investigation.

In this work, we synthesized the [229]Th-doped CaF$_2$ crystal through single crystal growth with a concentration of the order of $10^{18}$ cm$^{-3}$ (ref. 14). Subsequently, we excited the isomer state from the ground state of the doped [229]Th nucleus with a resonant X-ray beam, and observed the radiative decay to the ground state accompanied by the emission of a VUV photon. Using [229]Th doped crystals realizes a controlled chemical environment[14] in which the excitation to [229m]Th does not introduce disturbances due to radioactive decay or implantation of the parent isotopes of [229]Th which could promote non-radiative decay paths.

The closed-loop character of our experiment, with time-controlled isomer excitation followed by VUV-signal-detection, is a precursor for solid-state nuclear clocks using laser excitation and provides useful additional information for the excitation and decay dynamics of [229m]Th. One such important finding is the accelerated decay of the isomer state during X-ray beam irradiation, presenting a potential key process for quenching the remaining isomer population to the ground state: "re-initializing" the interrogation cycle for solid-state nuclear clock operation.

## Results

### Experimental overview

The experiment was performed at the BL19LXU beamline of SPring-8[18,19]. The data presented in this paper was taken in May and July 2023. The energy of the X-ray beam is tuned to the second excited state of the [229]Th nucleus at 29,189.9 eV (see Fig. 1a). The X-ray beam is monochromatized to an FWHM bandwidth of 30 meV by a Si monochromator system consisting of Si(111), Si(660) and Si(880). The beam has a spot size of ~$1.0 \times 0.5$ mm$^2$ at the target position with an intensity of $2 \times 10^{11}$ photons per second at maximum. The beam energy and intensity were regularly monitored by an absolute X-ray energy monitor[20] and free-air ionization chambers during the experiment. Before searching for the VUV-photon from [229m]Th decay using the VUV-detection system, the absolute center energy of the X-ray beam was confirmed at the NRS-detection system (see Fig. 1b) by detecting the nuclear resonant scattering (NRS) signal from the second excited state of [229]Th nuclei with a separately fabricated thorium nitrate target[21]. This resonance curve of the X-ray signal is obtained (Fig. 2a) by scanning the Si(660) and Si(880) monochromators. More information on the NRS measurement can be found in refs. 18, 22, 23.

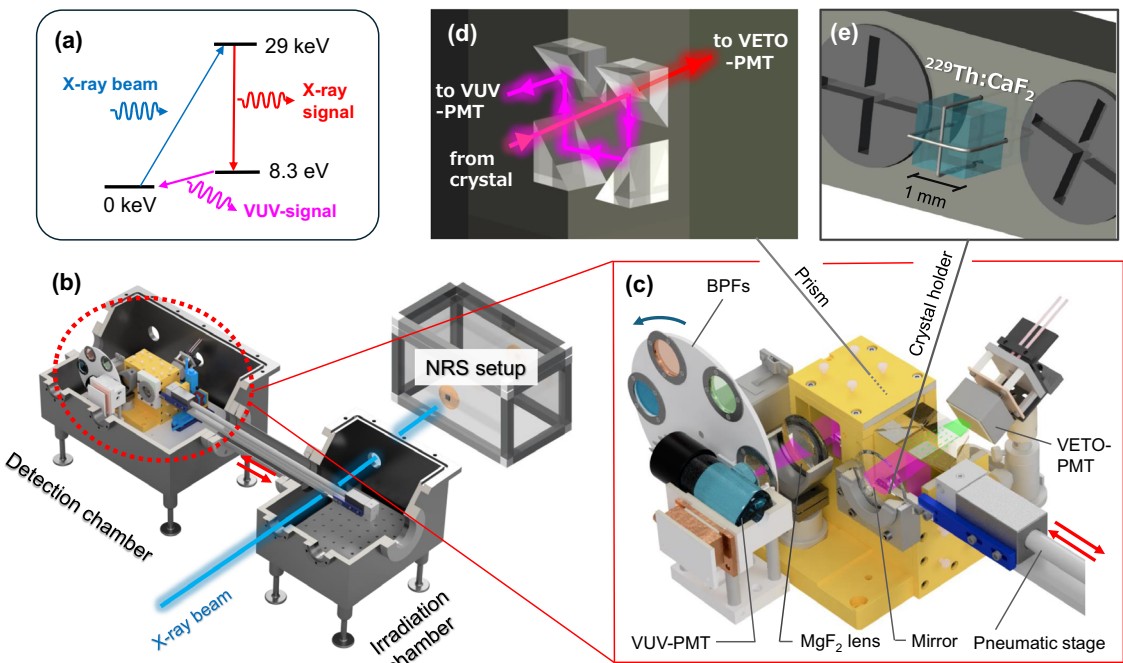

**Fig. 1 | Experimental setup. a** The nuclear levels of [229]Th and the scheme of isomer excitation and the observable signals (X-ray and VUV signals). **b** Overview of the system for isomer excitation and the detection of radiative decay. **c** Setup around the wavelength filtering devices inside the detection chamber; the moving target holder, the parabolic mirror, the prism set, the MgF$_2$ lens, the band-pass filters, the VUV-PMT, and the VETO-PMT. **d** The four consecutive prisms with dichroic mirror coating inside the wavelength filtering device. **e** Enlarged view of the thorium-doped crystal and the crystal holder. The crystal was fixed by using stainless wires.

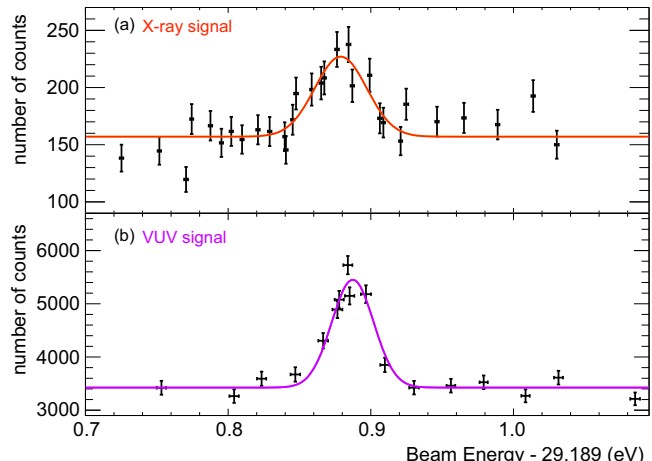

**Fig. 2 | Measured resonance spectra. a** The resonance spectrum of X-ray fluorescence (NRS signals) obtained with the thorium nitrate target. **b** The resonance spectrum of VUV-photon signals obtained with the $^{229}$Th:CaF$_2$ crystal. The horizontal axes of (**a**) and (**b**) indicate the absolute X-ray beam energy (offset by 29,189 eV). The vertical axis is the number of PMT counts in 1800 s after a beam irradiation time of ~600 s. The error bars in these plots represent the 68% confidence interval of statistical uncertainty. The background components (the constant offset in each Gaussian) show the beam-induced luminescence and radioluminescence from each target.

The $^{229}$Th-doped CaF$_2$ crystal targets were developed in the laboratory of TU-Wien[14] (see Methods). Two cuboidal pieces of $^{229}$Th:CaF$_2$ of appropriate size ~1 mm$^3$ (named "*Basso*" and "*Alto*") were cut from the same crystal ingot and facets were polished to optical quality. The measured $^{229}$Th activities were 20.0 kBq (17.2 kBq), yielding a $^{229}$Th doping concentration of $4.0 \times 10^{18}(4.7 \times 10^{18})$ cm$^{-3}$ for the "*Basso*" ("*Alto*") crystal, respectively. At such high concentrations, radiolysis during growth leads to a fluoride deficiency of the obtained crystal, reducing VUV transparency to <1% in the entire VUV region. Annealing the crystal under a fluoride-rich atmosphere recovered a transmission of >40% at 150 nm wavelength[24].

The experimental setup for VUV-photon detection consists of two connected vacuum chambers ("irradiation" and "measurement") with a vacuum level of 10$^{-3}$ Pa (see Fig. 1b). The target crystal is mounted on a moving holder (Fig. 1 (e)), which translates between the two chambers using a pneumatic stage. The crystal is first irradiated by the X-ray beam for a certain time to populate the $^{229m}$Th state in the irradiation chamber and is then moved to the measurement chamber within about one second. This transfer scheme is used to suppress X-ray beam-induced optical backgrounds.

The $^{229}$Th:CaF$_2$ crystal is positioned at the focal point of a parabolic mirror with a protected aluminum coating so that the emitted light is reflected and passes through a spectral filtering device as shown in Fig. 1c. The spectral filtering device consists of right-angle prisms with special dielectric coatings so that only light in a specific wavelength region around 150 nm is reflected. The use of four consecutive right-angle prisms (Fig. 1d) with installation directions orthogonal to each other guides the signal beam to the detector while reducing the number of background events down to $O(10^{-6})$ in the calculation. A set of selectable band pass filters (BPFs) installed on a rotating wheel is introduced for spectroscopic measurements of the radiation emitted by the crystal. The filtered light is focused by a MgF$_2$ lens and finally detected by a solar-blind photomultiplier tube (PMT, Hamamatsu R10454). The dark count rate of the PMT (called VUV-PMT) was reduced to less than 0.1 Hz by cooling it to −30 ℃.

Another PMT (Hamamatsu R11265-203, called VETO-PMT), which is sensitive to ultraviolet (UV) and visible (VIS) light, is installed behind the first right-angle prism to implement temporal filtering of the

optical signal (see Fig. 1b, c). The radioluminescence in the CaF$_2$ crystal caused by the occasional alpha- and beta-decay of $^{229}$Th and its daughter elements is known to produce bursts of order $O(10^4)$ photons per decay event, mainly in the UV region[25], which produces a background also on the VUV-PMT. Events detected by the VUV-PMT that coincide with such bursts detected by the VETO-PMT are rejected using a timing coincidence of the two PMTs.

The waveform signals from the PMTs are amplified and then recorded by an oscilloscope (National Instruments, PXIe-5162). See Methods for details of this signal processing. During the measurement, reference pulse signals are generated by a delay generator. They are combined with the VUV-PMT signal and used as the trigger channel. The reference pulse data are used to monitor the dead time of the data acquisition, and the remaining signal after background rejection.

## Analysis and results

The clear evidence for VUV photon emission from the $^{229}$Th isomer is the enhancement of the photon counts in the VUV-PMT when tuning the X-ray energy to be resonant with the 29-keV level. Figure 2b shows a resonance curve of the measured VUV-photon rate, using the Th:CaF$_2$ crystal and the VUV detection system. The crystal was irradiated for ~600 s for each X-ray energy point and VUV photons were counted for 1800 s after each X-ray irradiation. The resonance energy is obtained by fitting a Gaussian function plus a constant offset value. Taking into account the systematic uncertainties of the X-ray energy calibration[20], the peak energy is determined to be $E = 29{,}189.89 \pm 0.07$ eV. This energy coincides with that of the X-ray signal spectrum within the error, validating that the observed VUV signal is the result of isomeric decay.

The small difference in the resonance width ($\sigma = 18.2(25)$ and $13.4(11)$ meV in Fig. 2a and b, respectively) between these two resonance curves could be caused by the effects of the solid-state environment in the different targets: the resonance width is determined not only by the monochromator bandwidth but also by the target-dependent inelastic scattering processes corresponding to the phonon contributions in nuclear resonance scattering[26,27].

The wavelength of the VUV radiation emitted by the $^{229}$Th isomer was determined by measuring the transmission of this radiation through six different BPFs. The BPFs were switched every 10 s with a switching time of less than two seconds during the 1800s measurement period following an X-ray pumping isomer excitation. Measuring all six BPF transmissions during single excitation-decay sequence suppresses possible detrimental effects of beam intensity fluctuations and crystal damage on the wavelength measurement.

Figure 3 shows the determination process of the radiation wavelength from the thorium isomer. Each measured transmission $T_{meas}(i)$ ($i = 1-6$) was obtained (Fig. 3a) by taking the signal ratio with and without the filters. The filter number #0 corresponding to a blank one (without the filter) were used once in one rotating cycle for this $T_{meas}(i)$-calculation. The transmission spectra $T_{BPF}(i, \lambda)$ for the BPFs (Fig. 3b) were measured before and after the whole measurement campaign using a custom-built measurement system with a VUV-monochromator.

The wavelength of the emitted radiation is obtained by fitting the parameter $\lambda$ to minimize the weighted quadratic sum of the deviations between $T_{meas}(i)$ and $T_{BPF}(i, \lambda)$, describing as

$$\chi^2(\lambda) = \sum_{i=1}^{6} \left( \frac{T_{meas}(i) - T_{BPF}(i,\lambda)}{\Delta T_{meas}(i)} \right)^2. \quad (1)$$

From this procedure, the wavelength is determined to be $148.18 \pm 0.38$ (stat.) nm as shown in Fig. 3c, d. The systematic uncertainties for $T_{meas}(i)$ and $T_{BPF}(i, \lambda)$ were estimated by off-line systematic studies of BPF transmission, wavelength calibration, and their reproducibility (see Methods). Considering these systematic effects, the wavelength

of the $^{229}$Th isomer transition is determined as

$$\lambda = 148.18 \pm 0.38(\text{stat.}) \pm 0.19(\text{syst.}) \text{ nm}. \qquad (2)$$

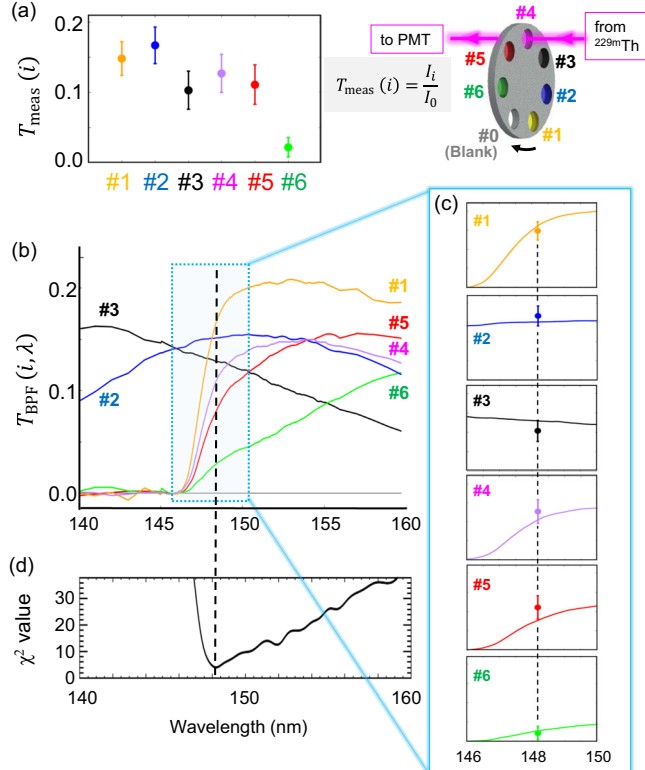

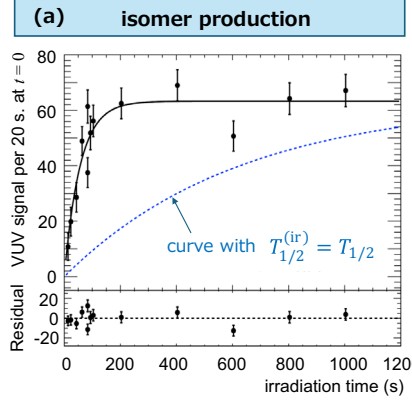

**Fig. 3 | Determination of the isomer transition wavelength. a** The measured transmissions of the radiation from the thorium isomer through the band-pass filters (BPFs), expressed as $T_{\text{meas}}(i)$ ($i = 1$–6). **b** The transmission spectra $T_{\text{BPF}}(i, \lambda)$ for each BPF. **c** The six overlaid plots between $T_{\text{BPF}}(i, \lambda)$ and $T_{\text{meas}}(i)$ for each BPF, where the $T_{\text{meas}}(i)$ are located at the best-fit wavelength (see the text). **d** The calculated $\chi^2$ distribution defined as Eq. (1). The error bars in (**a**) and (**c**) represent the 68% confidence interval of statistical uncertainty.

This value corresponds to an isomer energy of 8.367 ± 0.024 eV and is consistent with recently reported energies within the error[15,28–30].

The excitation dynamics can be observed by measuring the total isomer signal yield immediately after X-ray beam-off for various excitation periods (see Fig. 4a). The subsequent radiative decay dynamics can be directly extracted from PMT time traces recorded after termination of the excitation beam (see Fig. 4b). To eliminate effects of X-ray-induced crystal luminescence, the isomer signal is obtained by subtracting the temporal profile of the VUV signal after the excitation period at two different beam energies: one on the resonance peak of Fig. 2b and one far from (about 0.10 eV) the resonance peak, called "on-resonance" and "off-resonance", respectively.

Such a subtracted decay signal after the excitation period of the *Basso* crystal exhibits a clear exponential decay as expected from a nuclear process (Fig. 4b). The least squares fitting yields a half-life of $T_{1/2} = 431 \pm 23$ s. Taking into account several types of systematic uncertainty (see Methods), the result is $T_{1/2} = 431 \pm 23(\text{stat.}) \pm 26(\text{syst.})$ s.

The same experiment and analysis were performed using the *Alto* crystal to check consistency between samples with similar properties. The obtained half-life is $T_{1/2} = 459 \pm 21(\text{stat.}) \pm 23(\text{syst.})$ s, which is consistent with the *Basso*-crystal within the error. The half-life value from the combined data of both crystals is presented as the final result:

$$T_{1/2} = 447 \pm 16(\text{stat.}) \pm 20(\text{syst.}) \text{ s}. \qquad (3)$$

We observe no change in the measured isomer half-life using crystals with different doping concentrations, indicating that the effects of re-absorption of the emitted VUV isomer photons ("light-trapping") can be neglected (see Methods). We also observe no systematic change of the isomer decay half-life $T_{1/2}$ with parameters of the X-ray excitation beam; only the total amount of isomer state population is affected by the X-ray energy, photon flux, and spectral purity.

This is in strong contrast to the excitation dynamics, where we clearly observe a scaling of the isomer population timescale with X-ray flux. Figure 4 shows an excitation-decay dynamics for an X-ray beam flux of $2 \times 10^{11}$ photons/s.

The isomeric state population after the excitation time $T_{\text{ir}}$ is fitted by a build-up function $T_{1/2}^{(\text{ir})}/\ln 2 \cdot (1 - \exp(-T_{\text{ir}} \ln 2/T_{1/2}^{(\text{ir})}))$, with an independently introduced half-life $T_{1/2}^{(\text{ir})}$ during the beam irradiation.

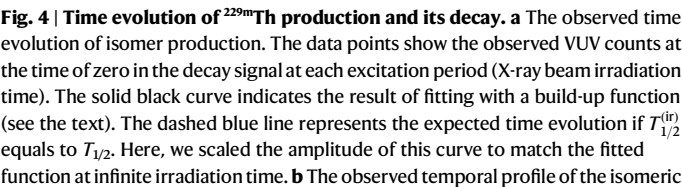

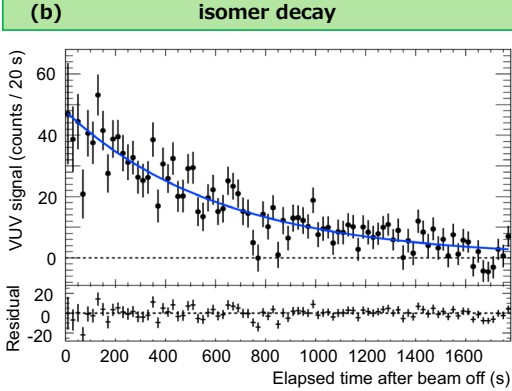

**Fig. 4 | Time evolution of $^{229\text{m}}$Th production and its decay. a** The observed time evolution of isomer production. The data points show the observed VUV counts at the time of zero in the decay signal at each excitation period (X-ray beam irradiation time). The solid black curve indicates the result of fitting with a build-up function (see the text). The dashed blue line represents the expected time evolution if $T_{1/2}^{(\text{ir})}$ equals to $T_{1/2}$. Here, we scaled the amplitude of this curve to match the fitted function at infinite irradiation time. **b** The observed temporal profile of the isomeric

decay signal after beam irradiation time of 600 s. The data points show the observed VUV counts at each elapsed time after the excitation beam was switched off. The blue solid line represents the result of fitting with an exponential decay function. The bottom row in (**a**) and (**b**) shows the corresponding residual plots for each fitting procedure. The error bars in these plots represent the 68% confidence interval of statistical uncertainty.

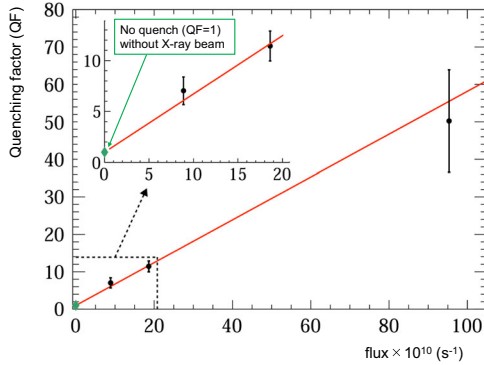

**Fig. 5 | Scaling of quenching factor $T_{1/2}/T_{1/2}^{(\text{ir})}$ with X-ray beam flux.** The horizontal axis indicates the weighted average beam intensity in photons per second. The inset shows an enlarged view near the zero-flux region. The green diamond-shaped point indicates the quenching factor at zero flux, which is equal to one. The point with the flux of $18.6 \times 10^{10}$/s corresponds to the excitation dynamics depicted in Fig. 4b. The error bars represent the 68% confidence interval of statistical uncertainty. The solid line indicates a fit with a linear function where the intercept is fixed at one.

We find an effective half-life

$$T_{1/2}^{(\text{ir})} = 39.2 \pm 4.9(\text{stat.}) \pm 5.8(\text{syst.}) \text{ s}, \qquad (4)$$

a factor of about ten times shorter than the isomer decay half-life $T_{1/2}$. This indicates the presence of an additional isomer decay channel during the exposure of the Th-doped crystal to the X-ray source, which we refer to as "X-ray quenching". We have investigated the scaling of this effect with X-ray intensity and found an approximately linear scaling of the quenching factor $T_{1/2}/T_{1/2}^{(\text{ir})}$ with X-ray beam flux as shown in Fig. 5.

These observations raise the question on the probability of emitting a VUV photon in a single decay event from the $^{229}$Th isomer back to the ground state (in the absence of the X-ray beam). The 8 eV nuclear-excited state in the crystal may interact with the surrounding chemical environment. As a result, the isomer state has the possibility to decay to the ground state through non-radiative processes.

The radiative transition probability $B_{\text{rad}}^{(\text{is})}$ is estimated by comparing the experimentally measured number of VUV-photons with the one calculated, based on the production yield of $^{229\text{m}}$Th and the experimental detection efficiency (see Methods).

For the parameters of Fig. 4, an excitation time of 600 s, an effective half-life $T_{1/2}^{(\text{ir})} = 39.2$ s, we expect to excite in total $(3.0–3.5) \times 10^5$ nuclei to the isomeric state which will decay back to the ground state with the half-life $T_{1/2} = 447$ s. Integrating the detected VUV count rate over 0–1800 s and dividing with the detection efficiency and trigger efficiency ($\epsilon_{\text{det}} = 6.5 \times 10^{-3}$ and $\epsilon_{\text{trig}} = 0.845$) (see Methods) yields a number of $3 \times 10^5$ emitters.

We conclude that the probability of radiative decay $B_{\text{rad}}^{(\text{is})}$ is consistent with one, the lower limit (90% confidence level) for this probability is estimated to be 0.45 (0.37) for the *Basso* (*Alto*) crystal, respectively.

## Discussion

Our findings indicate that, in the absence of the X-ray beam, radiative decay under the emission of a VUV photon is the dominant de-excitation process for the $^{229}$Th isomer in the crystal. However, more systematic studies are needed to clarify whether weak non-radiative decay channels exist and under which experimental conditions. The long half-life of $T_{1/2} = 447$ s measured after the excitation period is considered to be close to the half-life of the radiative decay, indicating the predicted narrow transition linewidth. Since the isomeric

transition is expected to be mainly magnetic dipole (M1)[31,32], the measured half-life in the crystal is corrected by applying a factor of $n^3$(ref. 16) to the half-life in vacuum if we assume the condition where internal conversion decay is forbidden. By using $n = 1.588$ at 148.2 nm for a CaF$_2$ crystal, the obtained half-life value for an isolated nucleus in vacuum is $1,790 \pm 64(\text{stat.}) \pm 80(\text{syst.})$ s, which is consistent with the recently reported isomer half-life of $1,400^{+600}_{-300}$ s of the trapped $^{229\text{m}}$Th$^{3+}$ ion[33].

This value is as long as expected by several theoretical works. In fact, there are several reports for the isomeric half-life of $O(10^3)$ s predicted using the microscopic nuclear model[31], and the simple rotational model of the deformed nucleus using the relevant experimental data[34,35]. More specific studies comparing predicted and measured values are expected in the future.

The half-life obtained from measurements recently performed at CERN-ISOLDE was 607(102) s[15]. Possible reasons for this discrepancy are the effect of different substrates (MgF$_2$ and CaF$_2$), different environmental conditions (heavy-ion implantation vs. crystal doping), and the effect of different depths of the Th-isomer end-position with respect to the crystal surface. In the present experiment, the pumped $^{229\text{m}}$Th nuclei are distributed over the entire (illuminated) crystal volume and are expected to interact with a common crystal environment (refractive index).

The significantly reduced (flux-dependent) isomer half-life observed during X-ray beam irradiation suggests the existence of additional decay paths. Possible candidate mechanisms are couplings to electronic near-bandgap or defect states, or electronic bridge processes in the electron cascades following X-ray-induced core-electron ionization. This "isomer-quenching" facilitates an artificial acceleration of nuclear isomeric decay, which has been discussed in other research fields, such as triggering the energy release of long-lived isomer states (sometimes called "isomer depletion")[36–38]. In atomic clock operation, the atomic state is initialized within milliseconds after measuring the excitation efficiency of the clock transition. However, there is currently no available method to initialize the isomer state to the ground within a short time scale for a solid-state nuclear clock. Identifying and controlling the present decay dynamics opens up the possibility to actively quench the remaining isomer populations, which can be used to speed up the interrogation sequences in future solid-state nuclear clocks based on $^{229}$Th[8].

A recent publication reported resonant laser excitation of the Thorium-229 isomer[39]. The excitation energy and radiative decay lifetime are compatible with values reported in this work. We find it interesting to note that the crystal X2 used in ref. 39 was cut from the same ingot as the "*Alto*" and "*Basso*" crystals used here.

In conclusion, we have excited the $^{229}$Th nuclear clock isomer from the ground state using resonant X-ray pumping and observed the subsequent VUV-photon emission of the radiative isomeric decay. We identify direct VUV photon emission as the predominant de-excitation process for $^{229}$Th doped into bulk CaF$_2$ single crystals. The determined isomer excitation wavelength is consistent with recently reported values[15,28–30]. We explored the temporal profiles of the excitation and decay dynamics and found substantially different half-lives, indicating qualitatively different coupling mechanisms between the isomer and the crystal environment. The X-ray-induced isomer quenching effect, as revealed in this study, is anticipated to become a pivotal element for the forthcoming operation of nuclear clocks.

## Methods

### Th-doped CaF$_2$ target

To obtain high doping concentrations exceeding $10^{18}$ cm$^{-3}$ with the available restricted amounts of $^{229}$Th source material, a miniaturized version to the commonly used vertical gradient freeze method was developed by TU-Wien with support by the Fraunhofer institute for integrated systems and device technology (IISB). We grow 3.2 mm

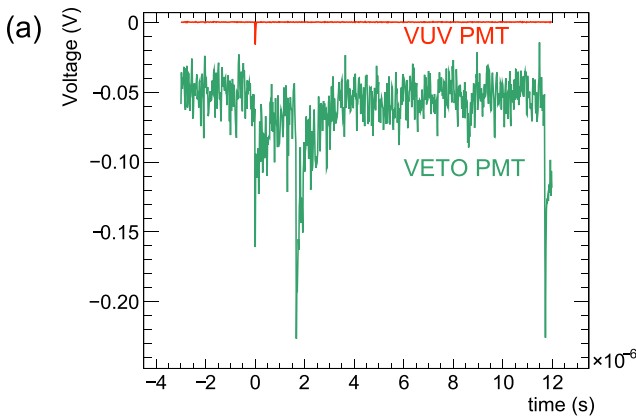

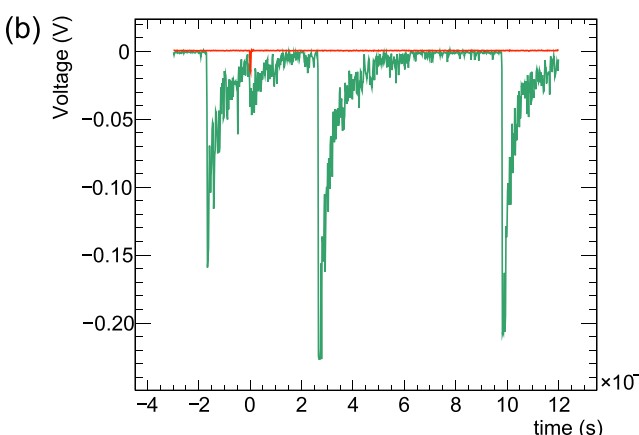

**Fig. 6 | An example of the waveform of the oscilloscope.** Time origin $t = 0$ indicates the timing of the triggered event from the VUV-PMT. Some strong peaks in the waveform of the VETO-PMT are caused by radioluminescence. **a** Waveform taken at the beginning of one measurement. **b** Waveform taken at the end of one measurement.

diameter, 11 mm long crystals by applying a steep temperature gradient (20 K/cm) across the starting growth material (dopant powder and single crystal seed) and dynamically control the liquid-solid interface. A typical growing speed is (<0.5 mm/h). The crystal growing device is kept under vacuum during the growth process, ~$10^{-4}$ mbar at the start of the growth. The vacuum prevents oxidation of the used graphite insulation and $CaF_2$ powder.

Crystal growing starts from a mechanically processed seed single crystals, which is milled down from commercially available bulk crystal (with known crystal orientation) to 3.2 mm cylinders of 11 mm height. From the top, a 2 mm diameter hole is drilled (5 mm) into the crystal to form a pocket for the doping material.

The doping material is 15 mg of $^{229}ThF_4$:$PbF_2$:$CaF_2$ powder. $PbF_2$ acts as a scavenger for oxygen removal and as a carrier that facilitates the handling of the minuscule (micrograms) amounts of $^{229}ThF_4$ during the wet chemistry preparation. $^{229}Th$ is obtained as dried nitrate from Oak Ridge National Laboratory, an activity of 1.6 MBq was used to grow the crystal used in the reported experiments. It is dissolved in 0.1 M $HNO_3$ and mixed with $PbF_2$ and $CaF_2$. Subsequently, $^{229}ThF_4$:$PbF_2$:$CaF_2$ was precipitated by addition of hydrofluoric acid, washed and dried (80 °C, 4 days).

The doping material is transferred into the pocket of the seed crystal using a funnel and inserted into the VGF growing device. Crystal growth itself consists of 5 sections: (1) 18 h of heating up the system to 800 °C, outgassing, and restoring pressure (2) 6 h of scavenging oxygen through reaction with $PbF_2$ at constant temperature (3) 22 h of melting the top half of the crystal and also slowly freezing it, reaching melting temperature around 1418 °C. The temperature (gradient) has

to be controlled to 1 °C (0.1 °C/cm) level, respectively, to adjust the melting depth into the crystal to 1 mm precision (4) 18 h of annealing the crystal at 1200 °C (5) 14 h of cooling down. A vacuum of at least $10^{-4}$ mbar is obtained before growth. During growth (especially during section 1) the pressure can go up to $10^{-2}$ mbar. The complete growth process typically takes 3 days.

After growth, the crystal is cut to the required dimensions using a diamond-wire saw (80 microns) and polished using standard polishing disks and slurry. Details of the crystal fabrication and characterization of the optical properties can be found in ref. 14.

To improve the VUV transmission, the crystal is heated again to ≈1150 °C in a separate device under $CF_4$ atmosphere at ambient pressure to counteract a fluoride deficiency that occurs due to radiolysis and losses during growth.

### DAQ scheme and background rejection

The data used in this work are derived from two voltage-time traces, simultaneously registered from the VUV-PMT and the VETO-PMT, recorded by an oscilloscope, that is triggered by an event on the VUV-PMT. During measurement, most of the triggering events are radioluminescence-induced background: 98–99.9% depending on the elapsed time from the stop of beam irradiation. Most of the radioluminescence background can be rejected using waveform information of the VETO-PMT. Basically, pulse height and timing information of radioluminescence are used for background rejection. An example of a waveform of radioluminescence background events is shown in Fig. 6. At the beginning of the measurement, strong photoluminescence signals were measured that caused a shift from the ground level. This shift size depends on the crystal used, the elapsed time from the stop of beam irradiation, and the reduction of the crystal transmittance due to beam irradiation. Therefore, the radioluminescence rejection condition depends on each measurement and the elapsed time in the measurement.

The threshold for the VUV-PMT signal was set to −5 mV, the minimum value at which it can operate effectively. For a quantitative estimate of the obtained VUV isomer photon yield, it is important to evaluate the data loss resulting from the failure to capture the distribution of signals with pulse height less than −5 mV.

To determine the trigger efficiency, an analysis was carried out on the decrease of the pulse-height distribution of the trigger signal from near-threshold to zero. By performing this analysis with different threshold settings, we estimated the number of low pulse-height signals (−10 to 0 mV) that were missed throughout the data acquisition (DAQ) procedure at the threshold of −5 mV. The efficiency was then estimated to be $\epsilon_{trig} = 84.5(14)\%$. The uncertainty in this estimation comes from the minimum edge value and the fitting error of the distribution function in the measured pulse height distribution.

The data acquisition efficiency $\epsilon_{DAQ}$ (or dead time) was also estimated because the oscilloscope may not record all triggered data due to accidental coincidence with another event or oscilloscope dead-time. The clock pulse (frequency of 10 Hz or 100 Hz depending on the run condition) of the delay generator was used as the additional trigger for the data recording in order to estimate the DAQ efficiency. The measured time-averaged efficiencies ranged from 98.31 to 99.95% in various runs.

Another important source of possible data loss is due to the rejection of the true VUV signal by the VETO-cut procedure. This VETO-cut efficiency $\epsilon_{veto}$ corresponding to the survival probability through the VETO-cut selection is estimated by the reduction of the clock trigger events by the VETO-cut selection. The efficiencies ranging from 27.0 to 32.7% were measured in different runs.

### Systematic uncertainties in lifetime measurements of $^{229m}Th$

As described in the main text, the half-life of the isomeric state after the X-ray beam termination was determined by fitting the temporal

spectrum (Fig. 4b) with the single exponential decay function. Three types of systematic uncertainty were considered in the analysis. One is the uncertainty in the constant offset parameter of the exponential decay function in the fitting procedure. This type of half-life uncertainty was evaluated to be 16 s in the analysis of the spectrum Fig. 4b. Another one is the uncertainty arising from the choice of cut parameters for pulse-height selection in the VETO-PMT signals. This uncertainty was estimated to be 12 s in the same spectrum. The last systematic uncertainty was estimated from the choice of the fitting time window. The possible maximum deviation was 16 s if we use the fitting time window 200–1780 s instead of 0–1800 s We also included the 16 s for the systematic error. By using the quadratic sum of the above three systematic uncertainties, the result of the half-life value for the *Basso* crystal is $T_{1/2} = 431 \pm 23(\text{stat.}) \pm 26(\text{syst.})$ s. The same evaluation of the systematic error was applied to other crystal data.

### Reabsorption effect

If the absorption length in the crystal of the on-resonance VUV-photon from the isomer state is smaller than the crystal length, so-called radiation trapping occurs due to reabsorption of the photon by the ground state of other $^{229}$Th nuclei[8]. Such a situation tends to prolong the observed half-life for the isomeric radiative decay. The efficiency of reabsorption (or reabsorption length) depends on the inhomogeneous broadening for the isomeric transition in the crystal, which is currently unknown. However, by comparing the measured radiative half-life of the isomer state for different $^{229}$Th doping concentrations, the effect of such reabsorption can be investigated. We additionally measured the isomer half-life with a low-density Th:CaF$_2$ crystal of $7 \times 10^{17}$ cm$^{-3}$, which is one order of magnitude lower concentration compared the *Basso/Alto*-crystals used in the main result. The measured half-life is $471 \pm 67(\text{stat.}) \pm 59(\text{syst.})$ s. This value is consistent with the measurement using the high $^{229}$Th density crystal (*Basso/Alto*). This means we did not observe any reabsorption effect in the radiative decay of $^{229\text{m}}$Th by other $^{229}$Th nuclei.

### Detection efficiencies

In the present scheme of the radiative decay measurement from $^{229\text{m}}$Th, the $^{229}$Th-doped CaF$_2$ target crystal is irradiated by the X-ray beam for a time period of $T_{\text{ir}}$. The number of $^{229}$Th isomer nuclei excited at the end of the excitation period $T_{\text{ir}}$ is calculated as

$$N_{\text{is}} = R_{\text{is}}\tau_{\text{ir}}\left(1 - \exp\left(-\frac{T_{\text{ir}}}{\tau_{\text{ir}}}\right)\right) \quad (5)$$

where $\tau_{\text{ir}}$ is the lifetime of the isomeric state inside the crystal target during beam irradiation ($\tau_{\text{ir}} = T_{1/2}^{(\text{ir})} / \ln 2$). $R_{\text{is}}$ is the production rate of $^{229\text{m}}$Th by the X-ray beam irradiation and is obtained by

$$R_{\text{is}} = R_{\text{2nd}}\text{Br}_{\text{tot}}^{\text{in}} \quad (6)$$

$$R_{\text{2nd}} = n_{\text{Th}}\sigma_{\text{eff}}\epsilon_{\text{shift}}\Phi l_{\text{x}}(1 - e^{-L/l_{\text{x}}}) \quad (7)$$

where $R_{\text{2nd}}$ is the excitation rate to the second excited state, Br$_{\text{tot}}^{\text{in}}$ is the branching ratio to the isomer state ("in"-band transition in the rotational band structure of the $^{229}$Th nucleus) including $\gamma$-decay and internal conversion. The parameter $\epsilon_{\text{shift}}$ indicates the reduction of excitation cross section due to a shift from the on-resonant energy. The parameters of $n_{\text{Th}}$, $\Phi$, $l_{\text{x}}$, and $L$ are the number column density of $^{229}$Th inside the crystal, the X-ray beam flux irradiating the crystals, the attenuation length of the X-ray beam in the crystal, and the crystal thickness, respectively. The $n_{\text{Th}}$ was estimated by measuring 193 keV $\gamma$-ray intensity following the $\alpha$-decay of $^{229}$Th by using a germanium detector. The effective cross section from the ground state to the

**Table 1 | The parameter values and their uncertainties for each component**

| Item | Value | Uncertainty |
|---|---|---|
| $\Gamma_\gamma^{\text{cross}}$ | 1.70 neV | 23.5% |
| $\lambda_{\text{2nd}}$ | 42.48 pm | 0.0003% |
| $\text{Br}_{\text{tot}}^{\text{in}}$ | 0.72 | 16.1% |
| $\sigma_{\text{x}}$ | 13.4 meV | 17.9% |
| $\Phi$ | $(1.48–1.86) \times 10^{11}$ s$^{-1}$ | 11.1% |
| $n_{\text{Th}}(Basso)$ | $4.0 \times 10^{18}$ cm$^{-3}$ | 6.8% |
| $n_{\text{Th}}(Alto)$ | $4.7 \times 10^{18}$ cm$^{-3}$ | 9.4% |
| $L(Basso)$ | 0.8 mm | 11.4% |
| $L(Alto)$ | 0.7 mm | 13.8% |
| $l_{\text{x}}$ @ 29.19 keV | 1.34 mm | 1% |
| $\tau_{\text{ir}}$ | 56 s | 13.4% |

The values with a range mean the run-dependent values. Some uncertainty parameters also depend on experimental conditions. These are used for the estimation of detectable VUV-photon number from the $^{229\text{m}}$Th.

second excited state of $^{229}$Th, $\sigma_{\text{eff}}$, is written as

$$\sigma_{\text{eff}} = \frac{\lambda_{\text{2nd}}^2}{4}\frac{\Gamma_\gamma^{\text{cr}}}{\sqrt{2\pi}\sigma_{\text{x}}} \quad (8)$$

where $\lambda_{\text{2nd}}$, $\Gamma_\gamma^{\text{cr}}$ are the wavelength and the cross-band transition width between the ground state and the second excited state of $^{229}$Th, respectively. The $\sigma_{\text{x}}$ is the root mean square energy width of the X-ray beam. The excitation width $\Gamma_\gamma^{\text{cr}}$ and its uncertainty are described in ref. 18. The branching ratio Br$_{\text{tot}}^{\text{in}}$ is also described in this article, but the updated value in the ref. 40 is used in the present analysis. The uncertainty of the branching ratio is estimated with the uncertainties of the relevant measured quantities. These parameter values and their uncertainties are summarized in Table 1.

Using the above estimated number of the isomer state population and several efficiency parameters, the detection rate of the VUV signal from the radiative isomer decay is calculated as follows:

$$R_{\text{det}}(t) = N_{\text{is}}B_{\text{rad}}^{(\text{is})} \cdot \epsilon_{\text{trig}}\epsilon_{\text{DAQ}}\epsilon_{\text{veto}}\epsilon_{\text{det}}C \cdot \frac{1}{\tau}\exp\left(-\frac{t}{\tau}\right) \quad (9)$$

where each parameter in this equation is explained in the following. The time origin $t = 0$ is defined as the timing when the target holder was moved in front of the focusing mirror. The parameter $\tau$ is the lifetime of the isomeric state inside the crystal in the absence of X-ray irradiation. The parameter $B_{\text{rad}}^{(\text{is})}$ is the probability of photon emission from $^{229\text{m}}$Th in the absence of X-ray irradiation, the efficiencies $\epsilon_{\text{trig}}$, $\epsilon_{\text{DAQ}}$ and $\epsilon_{\text{veto}}$ in the data processing are explained in section D.2. The parameter $\epsilon_{\text{det}}$ is the total detection efficiency of the experimental apparatus, which is written as a product of several component efficiencies as

$$\epsilon_{\text{det}} = \epsilon_{\text{crys}} \cdot \epsilon_{\text{geom}} \cdot \epsilon_{\text{mir}} \cdot \epsilon_{\text{prism}} \cdot \epsilon_{\text{lens}} \cdot \epsilon_{\text{PMT}} \quad (10)$$

where each term corresponds to the VUV-photon transmission of the crystal, geometrical efficiency, reflectance of the parabolic mirror, the reflectance of four dichroic mirror prisms, the transmission of the MgF$_2$ lens, and the quantum efficiency of the VUV-PMT, respectively. The $\epsilon_{\text{crys}}$, $\epsilon_{\text{mir}}$, $\epsilon_{\text{prism}}$, and $\epsilon_{\text{lens}}$ were measured by using a deuterium light source and a VUV spectrometer. The geometrical efficiency, which was estimated using optical simulations, depends on $\epsilon_{\text{crys}}$ and $\epsilon_{\text{lens}}$. Therefore, it is appropriate to evaluate these three detection efficiencies as a single term, which was estimated to be $\epsilon_{\text{crys}}\epsilon_{\text{geom}}\epsilon_{\text{lens}} = 0.071 \pm 0.019$. The remaining three efficiencies were estimated to be $\epsilon_{\text{mir}} = 0.89 \pm 0.04$, $\epsilon_{\text{prism}} = 0.45 \pm 0.02$, and

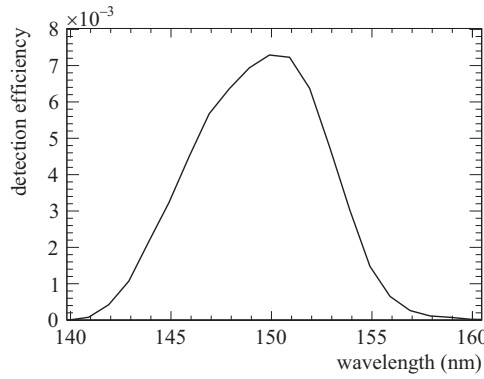

**Fig. 7 | The wavelength dependence of total detection efficiency of the isomeric radiation.** This total efficiency curve is estimated by multiplying the wavelength dependence of each efficiency in eq. (10).

**Table 2 | List of transition wavelengths used for calibration of the transmittance measurement system**

| Transition wavelength (nm) | Element |
|---|---|
| 130.2168 | O(I) |
| 130.4858 | O(I) |
| 130.6029 | O(I) |
| 141.1939 | N(I) |
| 149.2625 | N(I) |
| 149.2820 | N(I) |
| 149.4675 | N(I) |
| 174.2729 | N(I) |
| 174.5252 | N(I) |

The wavelength values are taken from NIST Atomic Spectra Database[41].

$\epsilon_{\mathrm{PMT}} = 0.22 \pm 0.01$. The parameter $C$ is the correction factor describing the reduction in crystal transmission by beam-induced damage and target holder movement time. The reduction factor due to crystal damage was estimated by observing the decrease in radioluminescence-induced VUV-event rates over several measurement cycles and scaling this decrease to an irradiation time of 600 s, resulting in a value of 0.997 per one excitation period of 600 s. The above efficiency parameters are estimated at the wavelength of 148 nm. The estimated wavelength dependence of the total detection efficiency $\epsilon_{\mathrm{det}}$ is plotted in Fig. 7.

**Wavelength determination**

The transmission spectra of each BPF and the dichroic-mirror set were measured with the custom-made setup consisting of a deuterium lamp (Heraeus, D200VUV), a VUV-monochromator (Vacuum & Optical Instruments, VMK-200-II), two collimators, and a photomultiplier (Hamamatsu, R6836). The systematic error in the wavelength determination arises from uncertainties in wavelength calibration for the monochromator. The absolute wavelength calibration of our assessment setup was performed using the discharge light source with the nitrogen and oxygen gas mixture. We used the several atomic transitions of O(I) and N(I) in the vicinity of 150 nm as listed in Table 2 for the calibration procedure. The uncertainty includes the error of the calibration itself and the effect of off-axis mounting of the light sources (deuterium lamp and $N_2/O_2$ gas discharge). The measured systematic uncertainty in the wavelength calibration was conservatively estimated to be 0.1 nm.

The wavelength of the radiation from the $^{229m}$Th was coarsely selected by the dichroic-mirror set placed upstream of the BPFs. This wavelength selection was investigated using different sets of

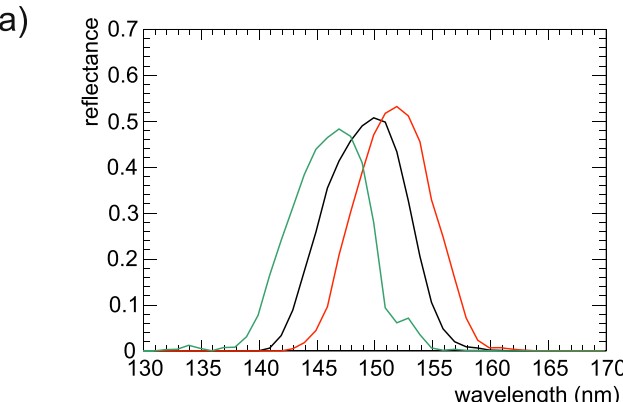

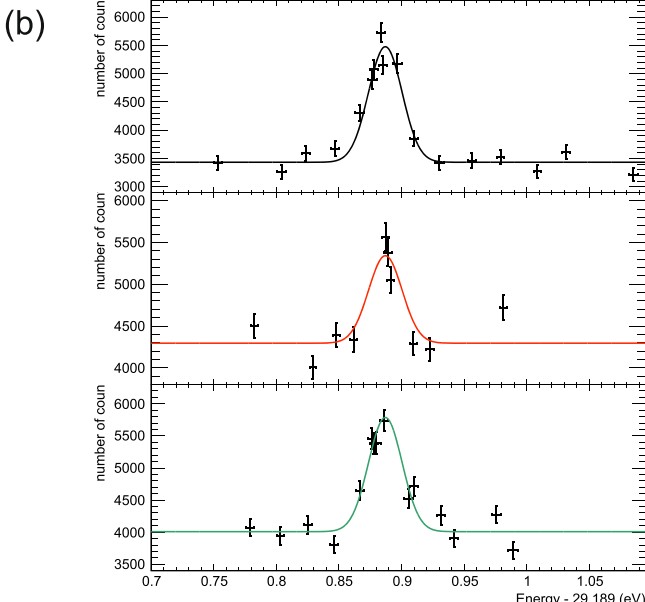

**Fig. 8 | Determination of wavelength using different dichroic mirrors.**
**a** Reflectance of different dichroic mirror sets used for determination of the wavelength of radiative decay photon from $^{229m}$Th. **b** The obtained energy spectra of VUV-signal with the three different dichroic mirror sets. Each point corresponds to a measurement result with a beam irradiation period of 600 s. The error bars represent the 68% confidence interval of statistical uncertainty. Solid lines show the simultaneous result of a Gaussian fit with a constant background of the three spectra, where center and width are the common parameters. Color of each line in (**a**) corresponds to that in (**b**).

dichroic mirrors without the BPFs. Figure 8a shows the reflectance of three different dichroic mirror sets. For each dichroic mirror set, the signal yield was measured by changing the X-ray beam energy. The wavelength $\lambda_{\mathrm{is}}$ is determined by minimizing the following $\chi^2$:

$$\chi^2(C, \lambda_{\mathrm{is}}) = \sum_{i=1}^{3} \left( \frac{N_{\mathrm{data}}(i)/\Phi(i) - C R(i, \lambda_{\mathrm{is}})}{\Delta(N_{\mathrm{data}}(i)/\Phi(i))} \right)^2, \qquad (11)$$

where $C$ and $\lambda_{\mathrm{is}}$ are the two independent variables for the minimization. The quantities $N_{\mathrm{data}}(i)/\Phi(i)$ is the number of events measured in the dichroic mirror set $i$, which is normalized by beam flux density $\Phi(i)$. $\Delta(N_{\mathrm{data}}(i)/\Phi(i))$ is its uncertainty. From this measurement, the wavelength is determined as

$$\lambda_{\mathrm{is}} = 148.14 \pm 0.40(\mathrm{stat.}) \pm 0.53(\mathrm{syst.}) \ \mathrm{nm}. \qquad (12)$$

This wavelength value is consistent with the $\lambda_{is}$ measurement using BPFs.

The wavelength of the radiation from the $^{229m}$Th is finally determined by comparing the transmission of this radiation through the six different BPFs and their transmission spectra as described in the main text. Therefore, the systematic uncertainties in wavelength determination come from both measurements of transmitted isomeric radiation and transmission spectra of BPFs.

One of the systematic uncertainties was determined by calculating the difference in wavelength between two spectra measured before and after the VUV search experiment. This uncertainty was found to be 0.05 nm. The uncertainty in the measurement of BPF-transmission spectra, which includes the position dependence inside each BPF and the measurement reproducibility, produces another systematic error in wavelength determination. This type of systematic uncertainty was measured to be 0.07 nm. Systematic uncertainty also comes from the analyzing procedure in the $T_{meas}(i)$-determination, such as the selection of a time window in the radiative decay signal and how to integrate the multiple measurement runs. These uncertainties were estimated to be 0.09 nm and 0.15 nm, respectively.

By taking the quadratic sum of the above several systematic errors, we concluded that the final result of the measured wavelength with the BPF system is

$$\lambda_{is} = 148.18 \pm 0.38 \text{(stat.)} \pm 0.19 \text{(syst.)} \text{ nm}, \tag{13}$$

as described in the main text.

## Data availability

All data needed to evaluate the conclusions in the paper are present in the paper. The data that generated in this study have been deposited in the Figshare repository https://figshare.com/s/5b3a21ce2a654b1db980.

## Code availability

The script to create some figures are available in the Figshare repository https://figshare.com/s/5b3a21ce2a654b1db980.

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

## Acknowledgements

The synchrotron radiation experiments were performed at the BL19LXU line of SPring-8 with the approval of the Japan Synchrotron Radiation Research Institute (JASRI) (proposals No. 2014B1524, 2015B1380, 2016A1420, 2016B1232, 2017B1335, 2018A1326, 2018B1436, 2019B1619, 2020A1284, 2021A1389, 2021B1516, 2022A1401, 2022A1732, 2022B1418, 2023A1358, 2023B1311) and RIKEN (No. 20180045, 20190051, 20200042, 20210042, 20220059, 20230042). The authors would like to thank all members of the SPring-8 operation and supporting teams. Special thanks should go to Mr. T. Kobayashi for technical assistance at SPring-8, to Dr. K. Konashi and Dr. M. Watanabe for $^{229}$Th extraction work, and to Mr. H. Kaino, Mr. K. Suzuki, Dr. H. Hara, Dr. Y. Miyamoto, and Dr. S. Uetake for support at Okayama. This work was supported by JSPS KAKENHI Grant Numbers JP18H01230, JP19K14740, JP19K21879, JP19H00685, JP21H01094, JP21H04473, JP22K20371, and JP23K13125. This work was also supported by JSPS Bilateral Joint Research Projects No. 120222003. T.H. acknowledges the Itoh Science Foundation for a grant. S.T. acknowledges the support by "Initiative for Realizing Diversity in the Research Environment" from MEXT, Japan, and Okayama University Dispatch Project for Female Faculties. T.M. is supported by the Inamori Foundation. This work has been funded by the European Research Council (ERC) under the European Union's Horizon 2020 research and innovation program [Grant Agreement No. 856415]. The project 23FUN03 HIOC [Grant DOI: 10.13039/100019599] has received funding from the European Partnership on Metrology, co-financed from the European Union's Horizon Europe Research and Innovation Programme and by the Participating States. This research was funded in whole or in part by the Austrian Science Fund (FWF) [Grant DOI: 10.55776/F1004, 10.55776/J4834, 10.55776/P33627, 10.55776/PIN9526523].

## Author contributions

T.H., K.O., Y.F., T.M., M.G., R.O., N.S., K.S., S.T., A. Yoshimi, K.Y., K.B., M.B., S.K., N.N., F.S., M.S., Y.S., K.T., A. Yamaguchi, and Y.Y. performed the synchrotron radiation experiments. Okayama University group developed the detector system. K.O. measured the performance of optical components. M.B., K.B., A.L., M. Pimon, M. Pressler, F.S., T. Schumm, T. Sikorsky developed the Thorium-229 doped crystals. Y.F., H.H., Y.K., K.O., Y.S., and K.Y. developed the thorium nitrate target for NRS measurements. H.F., T.W., and Y.Y. developed the absolute energy monitor. T.H. analyzed the data with input from all authors. T.H., T. Schumm, A. Yoshimi, and K.Y. prepared the manuscript with input from all authors. All authors discussed the results.

## Competing interests

The authors declare no competing interests.
