## [Peer Review File · Nature Communications]

Controlling ^{229}Th isomeric state population in a VUV transparent crystalREVIEWER COMMENTS

Reviewer #1 (Remarks to the Author):

Well written manuscript that describes a novel experiment with the goal to develop a novel solid-state based atomic clock using an nuclear isomer. The approach taken here is to excite ^{229}Th isomeric state through resonant X-rays and detect the photon emission. The detected photon energy is consistent with literature, however, new half-lives for excitation processes were found, pointing to a dependence of the values on the environment in which the nuclear is excited to the isomeric state.

The experiments are novel and well executed, and are highly relevant, both from a fundamental aspect but also for future application in solid-state based atomic clocks.

The team used well described and established experimental methods but applied them to a new field, the field of very-low energy nuclear metastable states. The details of the methods and analysis are well described and provide ample detail to support the conclusions. As a matter of transparency, the only request I have is to improve the figures 3 (a) by extending the shown time-lines to longer ranges (I would expect that in (a) some decay process could be visible, and for 3 (b) to include an additional panel that shows the residual fit difference. This is a method often used in precision (10^{-4} - 10^{-5} range) nuclear half-life measurements to indicate that there is no undetected systematic effect, or low-level contribution from other decays.

Reviewer #2 (Remarks to the Author):

Referee report.

Nature Communications manuscript NCOMMS-24-14525-T

Controlling ^{229}Th isomeric state population in a VUV transparent crystal

T. Hiraki et al

The manuscript by Hiraki and colleagues presents new results regarding the population and direct radiative detection of the low-lying isomeric state in ^{229}Th in a thorium-doped VUV transparent crystal, via resonant X-ray pumping at the Spring-8 synchrotron radiation facility, Japan. ^{229}Th has long been celebrated for hosting the lowest-lying isomeric state throughout the nuclear landscape, whose energy, determined in recent years using different techniques, is such that a future direct excitation from the nuclear ground state would be possible with lasers. In particular, and as noted by the authors in the manuscript and references, one of the outstanding opportunities lies in the potential for harnessing the isomer for an ultra-precise nuclear clock. Until very recently, direct evidence for the isomer was made through internal conversion studies, however in 2023 (Nature, Ref. 8) an important milestone was achieved as the radiative decay was detected for the first time.

The work presented here utilizes a very different approach to populate the isomeric state than Ref. 8, confirming the transition wavelength of the VUV radiative decay. The team utilize thorium-doped crystals ($^{229}\text{Th}:\text{CaF}_2$) prepared with a high concentration of source material (10^{18} ^{229}Th atoms/cm³). This follows the so-called solid-state approach to a future nuclear clock, with the crystal bandgap larger than the isomeric state energy, preventing internal conversion and thus facilitating observation of the radiative decay mechanism. The isomeric state itself is populated following irradiation of the crystal with a monochromatized X-ray beam, resonantly tuned to the second excited state of ^{229}Th (as indicated in Fig. 1). This state predominantly decays into the isomer via an internal conversion process, resulting in characteristic X-ray emission (similar to nuclear resonance scattering, NRS) that is used to determine the energy of the isomer. I note that this method of active pumping into the isomeric state was proven in pioneering measurements by this team in 2019 using a ^{229}Th target (Nature, Ref. 20). Here, the NRS setup (and target) remains, acting as a reference for the absolute center energy of the X-ray beam, with the $^{229}\text{Th}:\text{CaF}_2$ crystal installed along the X-ray beam path. A dedicated setup has been designed to detect the radiative decay photons while vetoing out the

overwhelming emission of radioluminescence events caused by the radioactive decay of the ^{229}Th in the crystal.

In addition to the determination of the isomer transition wavelength, the authors provide a measure of the decay half-life within the crystal and show that the probability of this decay mechanism is consistent with unity. Interestingly, the half-life value disagrees with that of Ref. 8, however plausible reasons are given. A key result which provides an important steppingstone for future clock operation is a new "X-ray quenching" effect. As highlighted in Fig. 3, the isomer decay is accelerated during the X-ray irradiation period, with a linear scaling as a function of X-ray flux that indicates the presence of an additional loss channel. Although this aspect is not yet understood and will need to be investigated further, it can be used to facilitate an artificial acceleration of the isomeric decay, allowing for a method of control that could be useful in the future.

Validation of the detection of the radiative photons is one of the most important aspects in this work. This is rigorously discussed in the Methods section of the article. The NRS signal, indicative of pumping into the isomeric state, provides a critical beam energy reference for the enhancement of VUV photons when on resonance (Fig. 1 f, g). The wavelength of the VUV radiation is measured using a series of bandpass filters, which in turn are calibrated using a custom-built VUV-monochromator. The data analysis procedure appears robust, with an extensive discussion of the systematic uncertainties for different aspects of the experiment (detector efficiency, data acquisition and the use of the veto detector for radioluminescence, lifetime measurement after different X-ray beam-off periods). To confirm the probability of radiative decay, the authors combine the measured number of VUV photons with an estimated number, requiring input from published theoretical parameters. I have not rigorously attempted to reproduce the numbers; however, the data has been carefully evaluated and the conclusions drawn are valid. The authors provide a convincing summary of the calculations. In summary, the article is well written and accessible to the general reader. The work has been put into context with recent high-impact results which have been correctly referenced. The experimental procedure and use of figures and tables is appropriate, and the data analysis procedure is valid, with critical detail in the Methods section given to the systematic error budget. Given the novelty of the methods presented to populate the isomeric state, as well as the detection procedure of VUV photons from the ^{229}Th -doped crystal, this is clearly an important milestone towards the solid-state approach being developed for a future nuclear clock. The controlled X-ray quenching mechanism is of particular interest as a possible means for future clock operation. It will be important to explore the physical mechanisms giving rise to this effect in subsequent measurements. These aspects of the work presented here clearly merit publication in this journal. I have only very minor suggestions for the authors to consider, detailed in the following.

1. I don't recall whether it was explicitly discussed, however it would be useful to comment on the background seen, particularly in the VUV signal, of Fig. 1. I understand that the 600 seconds of irradiation time ensures a saturation of the VUV signal as indicated in Fig. 3, with about 3 cts/s on resonance. When the X-ray energy is off-resonance, a background of about 3500 counts can be seen in the VUV spectrum – is this due to X-ray induced crystal luminescence, and does that have a decay timescale associated with it? I noticed this was alluded to on page 5, however it could be noted with respect to Fig. 1.
2. In the figure caption of Fig. 3b, the authors could add the irradiation time (600 s?).
3. Could the authors comment on whether any possible longer-term deterioration is expected or has been seen following X-ray irradiation? In the methods section D.5, there is a parameter C in Eq. 9 which represents a reduction factor due to crystal damage after an excitation period of 600s. Is this parameter sensitive to the irradiation time, or to the number of repeated irradiations? I wonder how it could have been determined to 0.997 given other potential uncertainties. The reason for the question is connected to the longer-term interest in controlling the depopulation of the isomer.

Reviewer #3 (Remarks to the Author):

This article reports a significant step forward in the development of a "nuclear clock" based on time-controlled excitation and de-excitation of the low-energy isomeric state in ^{229}Th . This general topic of the development of a nuclear clock using this transition has been a high priority for the community for more than a decade, with major significant advances in the past 5 years or so. The authors take advantage of the previously employed method of using high band-gap materials (in this case, a VUV transparent CaF_2 crystal) to sequester the ^{229}Th and populate the state through resonant X-ray pumping. Since the low-energy isomeric state (in its atomic configuration) decays almost exclusively by internal electron conversion, by placing it in a high band-gap material, this decay mode is highly suppressed through energy considerations, and thus the lifetime is dramatically extended since the photon emission mode dominates. The measured half-life reported here differs from the value reported previously, however the environment plays a major part in this, so I think there is physics here and not erroneous measurements.

The experimental work is extremely well done and presented at the right level. The manuscript was a joy to read, and I commend the authors for their work in preparing it well. The experiment is presented in enough detail that it can be reproduced (especially with the supplemental "methods" material), and uses clever experimental approaches from a variety of subdisciplines of science (as evidenced by the large author list as well). I do have some questions related to the analysis that may affect the results that the authors should evaluate. My main concern is with the treatment of the isomer decay data shown in Fig 3 b). Although the fit looks good "by eye", for half-life measurements in nuclear decay processes, a single exponential decay function is typically not suitable to perform precision half-life measurements at the few percent level, or better -- c.f. NIM A 579, 1005-1033 (2007) and citing articles. At the base level, the authors need to evaluate whether there are percent level contributions of backgrounds, other processes, or DAQ artifacts that could affect the single component fit. To first evaluate this, the authors should look at the fit residuals and examine any statistical trends. This should also include a multi-component fit that includes the possibility of at least one additional exponentially decaying component in the fit, to see if there are any unaccounted for statistical artifacts that could skew the final reported value.

From the physics discussion side, as I mention in the points below, the authors need to fix sections related to their description of the physical decay processes that cause the half-life differences between "in-medium" and "in vacuum". On the top of page 2 the authors discuss the environmental effects on the half-life, but need to start within the context of non-photon decay modes which are far more likely for neutral atoms in a vacuum at these extremely low energies (ie. to first order there are no photons emitted in the depopulation of ^{229m}Th in vacuum). This topic permeates its way through the document and into the conclusions as well.

There are a few corrections in the text:

1) On page 2, the authors report that the half-life in ^{229m}Th is shorter in a crystal than neutral atom in vacuum based on an index of refraction argument. This is incorrect, and quite the opposite is true. The measured vacuum half-life is $7(1) \times 10^{-6}$ s [PRL 118, 042501 (2017)] -- nearly 8 orders of magnitude shorter than the value reported here. This is due to the much more probable process of internal electron conversion (IC) which always competes with gamma decay in nuclear de-excitation. When the IC mode is forbidden or suppressed, such as in this case, the half life becomes long enough one can use ^{229m}Th for practical application, which is EXTREMELY sensitive to the environment the atom is subjected to (ie. MgF_2 or CaF_2) This section needs to be rewritten to reflect the correct physical processes.

2) Figure 1 is poorly laid out and difficult to follow. The caption of the figure (and accompanying text in the article body) are clear, but the figure negates all of this when referred to. I would either suggest breaking panels a,b,c,d into one figure separate from e,f,g in another. Space may limit this, but at the moment the figure detracts from the conceptual understanding of the experiment so this needs to be

addressed. I thought the other figures were well laid out.

3) The last paragraph on page 4 seems out of place and does not really make sense within the manuscript flow. I am not sure if this should be integrated into one of the others.

4) I think figure 4 should be removed or put in an appendix. This is discussed already in enough detail at the end of page 5 and I think it doesn't need to be included.

5) On page 6, there is again a discussion about isomeric decay half-lives "in vacuum" that needs to be corrected along the lines of what has been mentioned above. Further, the photon emission branch half-life is then compared to nuclear theory models in the text. The theory here is extremely challenging, and especially for the collective (rotational) models should not be expected to be correct, but are rather steps towards a future reliable prediction. I think this section is ok, but maybe remove the text about comparison to the theory and just state that theory models exist and are being refined, and compare to the previous experimental data.

Overall, the article is an important step in the development of a viable nuclear clock method, and with the above corrections and analysis investigations completed, I believe the article should be accepted for publication.

Point-by-point responses to reviewer #1

As a matter of transparency, the only request I have is to improve the figures 3 (a) by extending the shown time-lines to longer ranges (I would expect that in (a) some decay process could be visible, and for 3 (b) to include an additional panel that shows the residual fit difference. This is a method often used in precision (10^{-4} - 10^{-5} range) nuclear half-life measurements to indicate that there is no undetected systematic effect, or low-level contribution from other decays.

We agree with the reviewer that the Fig. 3(a) and (b) would not be fully satisfactory. We have included the residual plots for each fitting procedure in Fig.4 (redefined in the revised version) (a) and (b), and added the following sentence to the figure caption:

“The bottom row in (a) and (b) shows the corresponding residual plots for each fitting procedure.”

Regarding the request to extend the time line shown in Fig.3 (a) to longer ranges, we think it is not effective, because the plot in this figure already reaches the saturation value (equilibrium state between excitation and de-excitation of the isomer). To avoid misleading Fig.3 (a) and to make it clearer that its vertical axis indicates the signal amplitude at time zero in the decaying signal, we have changed the label of the vertical axis in redefined Fig.4(a) as following:

VUV signal (counts/20s) \rightarrow VUV signal per 20s. at $t = 0$

Furthermore, we have added more info about the vertical axis to the figure caption of redefined Fig.4(a):

“Here, we scaled the amplitude of this curve to match the fitted function at infinite irradiation time.”

Point-by-point responses to reviewer #2

1. I don't recall whether it was explicitly discussed, however it would be useful to comment on the background seen, particularly in the VUV signal, of Fig. 1. I understand that the 600 seconds of irradiation time ensures a saturation of the VUV signal as indicated in Fig. 3, with about 3 cts/s on resonance. When the X-ray energy is off-resonance, a background of about 3500 counts can be seen in the VUV spectrum – is this due to X-ray induced crystal luminescence, and does that have a decay timescale associated with it? I noticed this was alluded to on page 5, however it could be noted with respect to Fig. 1.

Thank you for the comments and suggestions. The answer to the reviewer's question is yes. The background level at the off-resonance region of 3,500 counts in the VUV signal is mainly due to X-ray induced crystal luminescence as described in page 5 with multiple decay-time components. The radioluminescence that could not be removed by the veto-cut procedure also contributes relatively small amount to this background level (see D.2: veto-cut efficiency). To make it clearer about this, we have made a comment on this background in the figure caption (re-defined as Fig.2), as

“The background components (the constant offset in each Gaussian) show the beam-induced luminescence and remained radioluminescence from each target.”

2. In the figure caption of Fig. 3b, the authors could add the irradiation time (600 s?).

Thank you for the comments. We have added information about the irradiation time of 600 s in the figure caption (re-defined Fig.4(b)) as

“The observed temporal profile of the isomeric decay signal.”

→

“The observed temporal profile of the isomeric decay signal after beam irradiation time of 600 s.”

3. Could the authors comment on whether any possible longer-term deterioration is expected or has been seen following X-ray irradiation? In the methods section D.5, there is a parameter C in Eq. 9 which represents a reduction factor due to crystal damage after an excitation period of 600s. Is this parameter sensitive to the irradiation time, or to the number of repeated irradiations? I wonder how it could have been determined to 0.997 given other potential uncertainties. The reason for the question is connected to the longer-term interest in controlling the depopulation of the isomer.

Thank you for important comment on the long-term deterioration of the target crystal due to X-ray beam irradiation. We estimated the reduction in VUV-light transmittance by observing the decrease in radioluminescence-induced VUV-event rates over several resonance-on/off measurement cycles. For example, we observed a 2.7% (typically a few % level, run-dependent value) reduction in the VUV-event over six-continuous measurements set (total irradiation time of $600 \text{ s} \times 6 \times 2(\text{on and off-resonance}) = 7,200 \text{ s.}$), leading to the reduction factor of 0.998 (again, run-dependent) for irradiation time of 600 s. We have averaged the run-dependent reduction factors and got the value of 0.997 in Appendix D.5. For recovery of transmittance, sometimes we replaced and annealed the irradiated crystal at 400°C during the experiment. We added more information for this explanation in D.5 as

Line 49-54 in page 12:

“The reduction factor due to the crystal damage was estimated to be 0.997 after an excitation period of 600 seconds.”

→

“The reduction factor due to crystal damage was estimated by observing the decrease in radioluminescence-induced VUV-event rates over several measurement cycles and scaling this decrease to an irradiation time of 600 s, resulting in a value of 0.997 per one excitation period of 600 s.”

Point-by-point responses to reviewer #3

I do have some questions related to the analysis that may affect the results that the authors should evaluate. My main concern is with the treatment of the isomer decay data shown in Fig 3 b). Although the fit looks good "by eye", for half-life measurements in nuclear decay processes, a single exponential decay function is typically not suitable to perform precision half-life measurements at the few percent level, or better -- c.f. NIM A 579, 1005-1033 (2007) and citing articles. At the base level, the authors need to evaluate whether there are percent level contributions of backgrounds, other processes, or DAQ artifacts that could affect the single component fit. To first evaluate this, the authors should look at the fit residuals and examine any statistical trends. This should also include a multi-component fit that includes the possibility of at least one additional exponentially decaying component in the fit, to see if there are any unaccounted for statistical artifacts that could skew the final reported value.

We thank the reviewer for careful reading and comment on our analysis for isomer decay spectrum. We agree to show how good the fit is for the decay spectrum. Thus, we have included the residual plot for the fitting procedure in Fig.4 (redefined in the revised version). As can be seen in this residual plot, there is no specific statistical trend. Furthermore, we have tried a multi-component (double component) exponential fit as the reviewer suggested and found no other time-constant component which indicates that the single exponential fit works well within the statistical error. Regarding the final half-life value, we investigated and adopted the several types of systematic errors as described in D.3 and D.4.

- 1) On page 2, the authors report that the half-life in ^{229m}Th is shorter in a crystal than neutral atom in vacuum based on an index of refraction argument. This is incorrect, and quite the opposite is true. The measured vacuum half-life is $7(1)\times 10^{-6}$ s [PRL 118, 042501 (2017)] -- nearly 8 orders of magnitude shorter than the value reported here. This is due to the much more probable process of internal electron conversion (IC) which always competes with gamma decay in nuclear de-excitation. When the IC mode is forbidden or suppressed, such as in this case, the half life becomes long enough one can use ^{229m}Th for practical application, which is EXTREMELY sensitive to the environment the atom is subjected to (ie. MgF_2 or CaF_2) This section needs to be rewritten to reflect the correct physical processes.

We thank the reviewer for pointing this out. We fully agree that the half-life of ^{229m}Th in vacuum becomes short (as short as $7\ \mu\text{s}$) when internal conversion decay is allowed. We would like to limit our description to the situation where internal conversion decay of ^{229m}Th is forbidden (as a ^{229}Th atom doped in CaF_2 crystal, or trapped $^{229}\text{Th}^{3+}$ ion (see ref.33)). To make our description

clearer, we have modified the sentence as follows:

Line 43-48 in page 2:

“In a crystal, the half-life of $^{229\text{m}}\text{Th}$ is expected to be reduced by a factor of n^3 (n being the refractive index of the crystal at the relevant wavelength) as compared to the vacuum half-life.”

→

If we restrict our discussion to cases where the charge state of thorium is such that it is $4+$, i.e. where internal conversion decay of $^{229\text{m}}\text{Th}$ is forbidden, the half-life of $^{229\text{m}}\text{Th}$ in a crystal is expected to be reduced by a factor n^3 (n being the refractive index of the crystal at the relevant wavelength) as compared to the vacuum half-life.

- 2) Figure 1 is poorly laid out and difficult to follow. The caption of the figure (and accompanying text in the article body) are clear, but the figure negates all of this when referred to. I would either suggest breaking panels a,b,c,d into one figure separate from e,f,g in another. Space may limit this, but at the moment the figure detracts from the conceptual understanding of the experiment so this needs to be addressed. I thought the other figures were well laid out.

Thank you for the comment and suggestion on Fig.1. We have split this figure into two parts as suggested by the reviewer. Here, the original (e) is included in the original (a)-(d) because the energy level diagram would be important in the beginning of this section. Therefore, we have redefined this energy level diagram as (a), included it in the setup parts of (b)-(e), and separated the observed resonance spectra into Fig.2 (a) and (b).

- 3) The last paragraph on page 4 seems out of place and does not really make sense within the manuscript flow. I am not sure if this should be integrated into one of the others.

Thank you for your comment and suggestions. We agree with this suggestion and remove the following sentence on the last paragraph on page 4.

“We investigated the time dynamics involved in the population and subsequent radiative decay of the isomeric state: clarifying how quickly the isomer is populated with the X-ray beam.”

→

(deleted)

- 4) I think figure 4 should be removed or put in an appendix. This is discussed already in enough

detail at the end of page 5 and I think it doesn't need to be included.

We agree that what Figure 4 (original) shows is well described in the text. On the other hand, it would also be useful to visualize data where this quenching factor is proportional to the beam flux. To make the significance of this clearer, we have included details of the region of the plot where the beam flux is close to zero (redefined Fig. 5). We have added the sentence for detailed information in the figure caption as

“The inset shows an enlarged view near the zero-flux region.”

- 5) On page 6, there is again a discussion about isomeric decay half-lives "in vacuum" that needs to be corrected along the lines of what has been mentioned above. Further, the photon emission branch half-life is then compared to nuclear theory models in the text. The theory here is extremely challenging, and especially for the collective (rotational) models should not be expected to be correct, but are rather steps towards a future reliable prediction. I think this section is ok, but maybe remove the text about comparison to the theory and just state that theory models exist and are being refined, and compare to the previous experimental data.

We thank the reviewer for the watchful comment on the discussion part. In the same way as in response to comment (1), we have modified the sentence as follows:

Line 96 in page 5 – Line 5 in page 6:

“Since the isomeric transition is expected to be mainly magnetic dipole (M1), the measured half-life in the crystal is corrected by applying a factor of n^3 ”

→

“Since the isomeric transition is expected to be mainly magnetic dipole (M1), the measured half-life in the crystal is corrected by applying a factor of n^3 to the half-life in vacuum if we assume the condition where internal conversion decay is forbidden.”

We also agree to remove the text about comparison between the theoretical and measured isomer half-life, because we do not like to discuss which theoretical values are closer to the measured values. Therefore, we have modified the relevant text as

Line 10-16 in page 6:

“This value is shorter than recently predicted theoretical values using the microscopic nuclear

model and the simple rotational model of the deformed nucleus using the recent experimental data. On the other hand, there are several evaluated half-life values using relevant experimental data comparable to our obtained values.”

→

“This value is as long as expected by several theoretical works. In fact, there are several reports for the isomeric half-life of $O(10^3)$ s predicted using the microscopic nuclear model, and the simple rotational model of the deformed nucleus using the relevant experimental data. More specific studies comparing predicted and measured values are expected in the future.”

REVIEWERS' COMMENTS

Reviewer #1 (Remarks to the Author):

I reviewed the updated manuscript and I am satisfied with the changes, I recommend publication as is, with the exception of the following suggestion: on page 6 the authors added a sentence with regards to the recent laser spectroscopy publication:

'After submission of this manuscript, resonant laser excitation of the Thorium-229 isomer was reported...' this sounds strange, considering that this publication will stand like this for years to come, and hence I would suggest to say something like: ' A recent publication of experiments using resonant laser excitation of the Thorium-229 isomer reported ...'

Reviewer #2 (Remarks to the Author):

In this revised version of the manuscript "Controlling 229Th isomeric state population in a VUV transparent crystal", the authors have satisfactorily addressed my few questions/comments. Additionally, the modifications made regarding the splitting of the original Fig. 1 into two separate figures adds to the readability and clarity of the manuscript. I noted the main text in this revised version has two minor corrections to be addressed (Page 4, line 19, and Page 5, line 25) in which pointers are made to the original figure (Fig. 1 f and g, respectively). These should now point to Fig. 2.

The addition of residual plots for the time evolution data in Fig. 4 supports the requirement of a single exponential decay curve in the fitting of the decay of the VUV fluorescence. Other rather minor additions to the text are noted.

As summarized in my previous report, this is an excellent and important contribution to the "roadmap" for the development of a clock based on the nuclear transition in 229Th. The timely nature of this article is emphasized by the additional reference added, acknowledging the first resonant laser excitation of the isomer, in which the resulting excitation energy and radiative decay lifetime are compatible with the values presented in this work.

I can recommend this article be accepted for publication.

Reviewer #3 (Remarks to the Author):

The revised version of the manuscript is significantly improved (particularly the figure), and the authors have addressed the referee comments in a suitable way. I recommend publication.

Point-by-point responses to reviewer #1

on page 6 the authors added a sentence with regards to the recent laser spectroscopy publication: 'After submission of this manuscript, resonant laser excitation of the Thorium-229 isomer was reported...' this sounds strange, considering that this publication will stand like this for years to come, and hence I would suggest to say something like: ' A recent publication of experiments using resonant laser excitation of the Thorium-229 isomer reported ...'

Thank you for the comment and the suggestion. We agree with the reviewer that this sentence should be corrected. Then, we have modified this sentence as follows:

Line 13-14 in page 5:

“A recent publication reported resonant laser excitation of the Thorium-229 isomer [ref.39].”

Point-by-point responses to reviewer #2

I noted the main text in this revised version has two minor corrections to be addressed (Page 4, line 19, and Page 5, line 25) in which pointers are made to the original figure (Fig. 1 f and g, respectively). These should now point to Fig. 2.

Thank you for the comments. We have corrected the sentence.

Line 48-49 in page 3:

“The small difference in the resonance width ($\sigma = 18.2(25)$, and $13.4(11)$ meV in Fig.2 (a) and (b), respectively) between these two resonance curves could be caused....”